# Gas Adsorption Response of Piezoelectrically Driven Microcantilever Beam Gas Sensors: Analytical, Numerical, and Experimental Characterizations

**DOI:** 10.3390/s23031093

**Published:** 2023-01-17

**Authors:** Lawrence Nsubuga, Lars Duggen, Tatiana Lisboa Marcondes, Simon Høegh, Fabian Lofink, Jana Meyer, Horst-Günter Rubahn, Roana de Oliveira Hansen

**Affiliations:** 1SDU NanoSYD, Mads Clausen Institute, University of Southern Denmark, Alsion 2, 6400 Sønderborg, Denmark; 2SDU Mechatronics, Department of Mechanical and Electrical Engineering, University of Southern Denmark, Alsion 2, 6400 Sønderborg, Denmark; 3AmiNIC ApS, Jernbanegade 75, 5500 Middlefart, Denmark; 4Fraunhofer Institute for Silicon Technology, Fraunhoferstraße 1, 25524 Itzehoe, Germany

**Keywords:** microcantilever, gas sensing, nonlinearity compensation, Euler–Bernoulli beam, Butterworth–Van Dyke, forced vibrations, polynomial fit, resonance frequency

## Abstract

This work presents an approach for the estimation of the adsorbed mass of 1,5-diaminopentane (cadaverine) on a functionalized piezoelectrically driven microcantilever (PD-MC) sensor, using a polynomial developed from the characterization of the resonance frequency response to the known added mass. This work supplements the previous studies we carried out on the development of an electronic nose for the measurement of cadaverine in meat and fish, as a determinant of its freshness. An analytical transverse vibration analysis of a chosen microcantilever beam with given dimensions and desired resonance frequency (>10 kHz) was conducted. Since the beam is considered stepped with both geometrical and material non-uniformity, a modal solution for stepped beams, extendable to clamped-free beams of any shape and structure, is derived and used for free and forced vibration analyses of the beam. The forced vibration analysis is then used for transformation to an equivalent electrical model, to address the fact that the microcantilever is both electronically actuated and read. An analytical resonance frequency response to the mass added is obtained by adding simulated masses to the free end of the beam. Experimental verification of the resonance frequency response is carried out, by applying known masses to the microcantilever while measuring the resonance frequency response using an impedance analyzer. The obtained response is then transformed into a resonance frequency to the added mass response polynomial using a polynomial fit. The resulting polynomial is then verified for performance using different masses of cantilever functionalization solution. The functionalized cantilever is then exposed to different concentrations of cadaverine while measuring the resonance frequency and mass of cadaverine adsorbed estimated using the previously obtained polynomial. The result is that there is the possibility of using this approach to estimate the mass of cadaverine gas adsorbed on a functionalized microcantilever, but the effectiveness of this approach is highly dependent on the known masses used for the development of the response polynomial model.

## 1. Introduction

Piezoelectrically driven microcantilevers (PD-MCs) have been widely used in gas sensing applications through the adsorption of target gas molecules on functionalized surfaces [1,2,3]. The adsorption of target gas molecules causes a change in the surface stress of the functionalized microcantilever, resulting in a change in resonance frequency and deflection [4,5,6,7]. We previously studied the application of PD-MCs in the electronic nose for the detection of 1,5-diaminopentane(Cadaverine), as a measure of the extent of meat spoilage [8,9]. Cadaverine is a biogenic amine formed by the decarboxylation of amino acids by enzymes of bacterial origin [10], whose concentration has been observed to increase during the spoilage of meat [11]. Therefore, the presence of few target gas molecules is reflected by changes in the resonance frequencies being measured and impedance output actuated through piezoelectric material [12].

Sensing and detection are typically performed using two approaches, the dynamic approach and the static approach. The dynamic approach relies on the analysis of change in resonance frequency while the static approach uses the change in deflection of the microcantilever. In both cases, the change in property measured is proportional to the mass of adsorbed [13,14,15]. The dynamic approach has recently been favored over the static approach due to varying complexities in methods used in the measurement of the deflection of microcantilever sensors. Optical deflection measurement is the most employed method for static approach and has been and has been reported to come with complications in readout and complexity in set-up and alignment, affecting the sensitivity [16,17]. The dynamic approach, on the other hand, offers other readout parameters besides changes in resonance frequency. These include the piezoresistive readout, where the change in resistance of the microcantilever is measured using a Wheatstone bridge with three reference resistors, of which one is adjustable [18], the piezoelectric readout where the piezoelectric microcantilever is driven with the inverse piezoelectric effect (self-excitation) by applying an electric AC voltage to the piezoelectric material, and sensing microcantilever bending by recording the piezoelectric current change [4,19]. Previous studies on gas-sensing applications using PD-MCs have employed the dynamic approach, where the microcantilever is actuated by applying a sinusoidal voltage at a frequency near its natural frequency, to a piezoelectric layer located at the clamped end [20,21,22].

While the PD-MC-based sensors are considered to perform generally well, they have challenges that affect the quality of the measurements. The piezoelectric material layer provides non-linear responses originating from non-linear elastic vibrations [23], and forced vibrations [24]. Other notable sources of nonlinearity and noise in the response signal include flexural rigidity of the beam [25] and adsorption surface-induced stresses [4,26], changes in temperature causing a thermal drift in the output signal, noise in the driving voltage, and surface stress changes originating from the functionalization surface-coating modification due to target molecular binding and unbinding [27,28,29,30,31]. It has been vastly documented that the resonance frequency of a microcantilever beam varies with mass added by using a relationship that can be summarized as [32,33,34]:(1)$$f_{m}=\frac{1}{2\pi }\sqrt{\frac{k}{m+\delta_{m}}}$$
where $f_{m}$ is the resonance frequency, *k* is the lumped stiffness of the microcantilever system, *m* is the lumped mass, and $\delta_{m}$ is the added mass. The major drawback of this relationship is that it assumes constant stiffness *k*, neglecting the effects of changes to surface stress and subsequently *k*.

Various analytical studies on mass sensing using dynamically actuated PD-MCs have been conducted based on the relationship in Equation (Equation 1). Bouchala et al. (2017) analytically investigated the linear dynamic response of beams to added mass modeled as discrete masses. The study found that the nonlinearity in the response signal was introduced by the electrical excitation when the microcantilever was dynamically excited [35]. Lam et al. (2022) further investigated the effect of driving a MEMS resonator subjected to an external dynamic load, with a signal buried in a multi-frequency noise floor. Despite the findings of improved signal identification when the system is actively driven near its natural frequency, there was a result showing noise in the output signal, requiring post-processing for improved signal-to-noise ratios [36]. Qiao et al. (2022) analyzed the response to thermal noise in electrostatic microcantilever MEMS sensors and found that mechanical–thermal and electrical–thermal noise dominated the output signal, especially when the stability of co-existing system motions were significantly smaller than the noise-driven motions [37].

Various strategies to address the nonlinearity and noise in dynamically actuated PD-MCs have been attempted, with particular emphasis on the development of mathematical response process models for nonlinearity compensation [38,39,40,41] and microcantilever non-linear vibration control [42,43]. However, they all rely on the determination of corrective system parameters during calibration, which are kept constant over time. This makes them unsuitable for systems whose mechanical characteristics—such as rigidity and Young’s modulus of the functionalization layer—evolve with usage. Alternative approaches (that have been advanced by some studies) are focused on directly addressing nonlinearity in PD-MCs response signals from specific sources, including the development of mathematical response models to address the fluctuations in PD-MC response signals originating from molecular binding and unbinding interactions with functionalized surfaces [44]. The rigidity of adsorbates on the functionalized surface has also been studied, resulting in a method to accurately determine the adsorbed mass, inhomogeneously distributed over the cantilever surface [45]. A simulation model for adsorption-induced stress due to molecular adsorption on the microcantilever surface has also been developed, taking into account the inter-molecular interactions between adsorbed atoms and functionalized surfaces [46]. The predicted deflection from the stress affects resonance frequency and can be used to accurately estimate the adsorbed mass [47]. Research on the response of dynamically actuated PD-MCs has also revealed that fluctuations in the response signal are caused by the noise present in the actuating and readout equipment used [48]. Although the studies enumerated have provided models for correcting the nonlinearity in PD-MC-based sensor response signals and improving on adsorbed mass estimation, they are all modeled around structural parameters pertinent to the PD-MC system and neglect contributions from actuating and readout equipment.

This study aims to demonstrate that by parameterizing the resonance frequency response to the mass added of a complete piezoelectrically driven microcantilever-based sensor system into a polynomial of the form:(2)$$F_{M}=\alpha M^{3}+\beta M^{2}+\gamma M+\lambda$$
where *M* is the total mass of the PD-MC, $F_{m}$ is the resonance frequency response from the system, and α, β, γ, and λ are polynomial coefficients determined experimentally, it is possible to estimate the mass of adsorbed target gas. It is our hypothesis the resulting polynomial is a true response of the system with all source response signal disturbances accounted for. Therefore, instead of modeling all of the possible sources of noise and nonlinearity, the response of the PD-MC sensor system to known masses can be used to develop a mathematical response model on which the estimation of the mass of the target gas molecules adsorbed could be based. We limit our study to the adsorption of 1,5-diaminopentane(Cadaverine) on a functionalized PD-MC, based on our previous studies where a microcantilever-based sensor for meat and fish freshness was developed [8,9].

Subsequent parts of this article are organized as follows: In Section 2, we carry out a free and forced numerical vibration analysis for a chosen cantilever design with desired resonance frequency. In this analysis, we derive general PDEs for a stepped beam, which can be extended to beams or arbitrary shapes and sizes. We then transform the resultant model into an equivalent electrical model and simulate resonance frequency to the added mass response. The section is concluded with experiments on the adsorption of cadaverine using a fabricated microcantilever. In Section 3, we present the results obtained from the numerical and experimental studies, and we discuss them in Section 4. In Section 5, we present our conclusions and recommend ideas for future research.

## 2. Materials and Methods

In order to parameterize the non-linear cantilever response to adsorbed mass, we used an analytical approach to determine the free and forced vibration resonance frequencies and corresponding mode shapes and verified the model using the finite element analysis in ANSYS. The forced microcantilever vibrations due to piezo-actuation were then analytically evaluated, and the consequent model was transformed into an equivalent electrical circuit based on the Butterworth–Van-Dyke (BVD) model [49], by evaluating the impedance and resonance frequency response for different masses added to the system. The experimental verification of the cantilever resonance frequency to adsorbed gas mass was then conducted. The obtained response profile was then transformed into a polynomial, to form a basis for estimating unknown masses.

### 2.1. Analytical Approach

Figure 1 shows the chosen cantilever baseline. The cantilever beam material is Poly-silicon with a piezoelectric material (AlN) deposited and tightly bonded on the fixed end of the microcantilever. This creates jumped non-uniformities in terms of geometrical structure and materials, making the uniform beam model inadequate for the analysis. The beam is therefore divided into *N* segments. Appropriate governing conditions are assumed with constant parameters for respective sections, with continuity conditions at stepped points.

#### 2.1.1. Transverse Vibration Analysis

The analytical model used assumes that the piezoelectric-driven cantilever obeys the Euler–Bernoulli (EB) beam theory, where the thickness of the beam is considered very small compared to its length *L*. The microcantilever beam is considered to have a total length *L*, width wb, and thickness tb. The piezoelectric patch is of the length $l_{1}$, width wp, and thickness tp. *E* and *I* denote Young’s modulus and variable moment of inertia, respectively. From this point, henceforth, subscripts *p* and *b* indicate the designation of the property to the piezoelectric layer or elastic beam layer, respectively, and *n* denotes the numbering of the section under consideration.

The transverse vibration equation of the beam is derived using Hamilton’s principle. Small transverse deflections w(x,t) are considered at any point *x* along the beam, to allow for the assumption of linear properties on the system [50,51]. Assuming that a plane section of the beam remains plane after deformation by axial displacement *z* from the neutral axis, the axial strain can be expressed as
(3)$$\varepsilon_{x}=-z\frac{d^{2}w}{dx^{2}}$$
and the axial stress as
(4)$$\sigma_{x}=E\varepsilon_{x}$$

Considering the section of the beam with the piezoelectric patch, the neutral axis is shifted. The strain in that section is modified and can be expressed as:(5)$$\varepsilon_{x}=-\left(y-y_{n}\right)\frac{d^{2}w\left(x,t\right)}{dx^{2}}$$
where $y_{n}$ is the term connoting the shift in the neutral axis and can be defined as:(6)$$y_{n}=\frac{1}{2}\left(\frac{E_{p}t_{p}\left(t_{p}+t_{b}\right)}{E_{p}tp+E_{b}t_{b}}\right)$$

The strain energy of the beam π is expressed as:(7)$$\pi =\int_{0}^{L}\frac{d^{2}}{dx^{2}}\left[\left(EI\right)\left(x\right)\frac{d^{2}w\left(x,t\right)}{dx^{2}}\right]dx$$

Since the beam is characterized by material and structural discontinuities along its length, the variable stiffness and the moment of inertia are expressed as:(8)$$\begin{array}{c} EI\left(x\right)=E_{b}I_{b}\left(x\right)+E_{p}I_{p}\left(x\right)\\ I_{b}\left(x\right)=\frac{1}{12}bt_{p}^{3}+H\left(x\right)bt_{b}y_{n}^{2}\\ I_{p}\left(x\right)=\left[\frac{1}{12}bt_{p}^{3}+bt_{p}y_{n}^{2}{\left(\frac{1}{2}\left(t_{b}+t_{p}\right)-y_{n}\right)}^{2}\right]bH\left(x\right) \end{array}$$

The work of the driving force $F_{p0}$ is given by
(9)$$\delta W\left(t\right)=F_{p0}V\left(t\right)\int_{0}^{l_{1}}\frac{d^{2}H\left(x\right)}{dx^{2}}dx$$
where $F_{p0}$ defines the strain developed from the application of an electric field along the electrodes of the piezoelectric element patch and is defined as
(10)$$F_{p0}=bE_{p}d_{31}\left[\frac{1}{2}\left(t_{b}+t_{p}\right)-y_{n}\right]$$
where H(x) is the heaviside function.

The kinetic energy KE is written as
(11)$$KE=\frac{1}{2}\int_{0}^{L}\rho \left(x\right)A\left(x\right){\left[\frac{dw(x,t)}{dx}\right]}^{2}dx$$
where ρ and *A* are the varying density and area of the cantilever beam, respectively. Using the Hamiltonian principle, we consider the total energy experienced by the vibrating system, which is expressed as:(12)$$\frac{d^{2}}{dx^{2}}\left(E\left(x\right)I\left(x\right)\frac{d^{2}}{dx^{2}}w\left(x,t\right)\right)+m\left(x\right)\frac{d^{2}}{dt^{2}}w\left(x,t\right)=F_{p0}V\left(t\right)\int_{0}^{l_{1}}\frac{d^{2}H\left(x\right)}{dx^{2}}dx$$

Considering free vibrations, Equation (Equation 12) is evaluated to
(13)$$\frac{d^{2}}{dx^{2}}\left(E\left(x\right)I\left(x\right)\frac{d^{2}}{dx^{2}}w\left(x,t\right)\right)+m\left(x\right)\frac{d^{2}}{dt^{2}}w\left(x,t\right)=0$$

Equation (Equation 13) can be separated into its spatial and temporal components resulting in
(14)$$\frac{d^{2}}{dx^{2}}\left(E\left(x\right)I\left(x\right)\frac{d^{2}}{dx^{2}}w\left(x,t\right)\right)/\left(m\left(x\right)\phi \left(x\right)\right)=-\ddot{q}/q\left(t\right)=\omega^{2}$$
(15)$$\frac{d^{2}}{dx^{2}}\left(E\left(x\right)I\left(x\right)\frac{d^{2}}{dx^{2}}w\left(x,t\right)\right)=\omega^{2}m\left(x\right)\phi \left(x\right)$$
where ω is the natural frequency in radians and can explicitly be expressed using $\beta_{n}$ as a function of resonance frequency as
(16)$$\omega^{2}={\left(\beta_{n}\right)}^{4}\frac{{\left(EI\right)}_{n}}{m_{n}}$$

Equation (Equation 15) can further be simplified into
(17)$$\frac{d^{4}}{dx^{4}}\phi_{n}\left(x\right)-\beta_{n}^{4}\phi_{n}\left(x\right)=0$$

With the general solution:(18)$$\phi_{n}\left(x\right)=A_{n}sin\beta_{n}\left(x\right)+B_{n}cos\beta_{n}\left(x\right)+C_{n}sinh\beta_{n}\left(x\right)+D_{n}cosh\beta_{n}\left(x\right)$$
where $\phi_{n}\left(x\right)$ is the solution to the mode shape function at a given point *x*. By applying appropriate boundary conditions for a clamped-free beam on the first section (fixed end of the beam) (Equation (Equation 19)), solutions to the mode shape equation coefficients are obtained using Equation (Equation 20).
(19)$$\begin{array}{c} \phi_{1}\left(0\right)=\frac{d\phi_{1}\left(0\right)}{dx}=0 \end{array}$$
(20)$$B_{1}+D_{1}=0,A_{1}+C_{1}=0$$

By applying continuity equations at discontinuity points, we obtain mode shape coefficients for each $n^{th}$ section using Equations (Equation 21) and (Equation 22).
(21)$$\begin{array}{c} \phi_{n}\left(l_{n}\right)=\phi_{n+1}\left(l_{n}\right)\\ \frac{d\phi_{n}\left(l_{n}\right)}{dx}=\frac{d\phi_{n+1}\left(l_{n}\right)}{dx}\\ {\left(EI\right)}_{n}\frac{d^{2}\phi_{n}\left(l_{n}\right)}{dx^{2}}={\left(EI\right)}_{n+1}\frac{d^{2}\phi_{n+1}\left(l_{n}\right)}{dx^{2}}\\ {\left(EI\right)}_{n}\frac{d^{3}\phi_{n}\left(l_{n}\right)}{dx^{3}}={\left(EI\right)}_{n+1}\frac{d^{3}\phi_{n+1}\left(l_{n}\right)}{dx^{3}} \end{array}$$
(22)$$\begin{array}{c} A_{n}sin\beta_{n}l_{n}+B_{n}cos\beta_{n}l_{n}+C_{n}sinh\beta_{n}l_{n}+D_{n}cosh\beta_{n}l_{n}=A_{k}sin\beta_{k}l_{n}+B_{k}cos\beta_{k}l_{n}+C_{k}sinh\beta_{k}l_{n}+D_{k}cosh\beta_{n+1}l_{n}\\ A_{n}cos\beta_{n}l_{n}-B_{n}sin\beta_{n}l_{n}+C_{n}cosh\beta_{n}l_{n}+D_{n}sinh\beta_{n}l_{n}=A_{k}cos\beta_{k}l_{n}-B_{k}sin\beta_{k}l_{n}+C_{k}cosh\beta_{k}l_{n}+D_{k}sinh\beta_{n+1}l_{n}\\ {\left(EI\right)}_{n}\beta_{n}^{2}\left(-A_{n}sin\beta_{n}l_{n}-B_{n}cos\beta_{n}l_{n}+C_{n}sinh\beta_{n}l_{n}+D_{n}cosh\beta_{n}l_{n}\right)={\left(EI\right)}_{k}\beta_{k}^{2}\left(-A_{k}sin\beta_{k}l_{n}-B_{k}cos\beta_{k}l_{n}+C_{k}sinh\beta_{k}l_{n}+D_{k}cosh\beta_{k}l_{n}\right)\\ {\left(EI\right)}_{n}\beta_{n}^{3}\left(-A_{n}cos\beta_{n}l_{n}+B_{n}sin\beta_{n}l_{n}+C_{n}cosh\beta_{n}l_{n}+D_{n}sinh\beta_{n}l_{n}\right)={\left(EI\right)}_{k}\beta_{k}^{3}\left(-A_{k}cos\beta_{k}l_{n}+B_{k}sin\beta_{k}l_{n}+C_{k}cosh\beta_{k}l_{n}+D_{k}sinh\beta_{k}l_{n}\right) \end{array}$$
where k=n+1. Finally at the free end (x=L), the boundary conditions are:(23)$$\frac{d^{2}\phi_{N}\left(l_{N}\right)}{dx^{2}}=\frac{d^{3}\phi_{N}\left(l_{N}\right)}{dx^{3}}=0$$

Resulting in
(24)$$\begin{array}{c} \beta_{N}^{2}\left(-A_{N}sin\beta_{N}l_{N}-B_{N}cos\beta_{N}l_{N}+C_{N}sinh\beta_{N}l_{N}+D_{N}cosh\beta_{N}l_{N}\right)=0\\ \beta_{N}^{3}\left(-A_{N}sin\beta_{N}l_{N}-B_{N}cos\beta_{N}l_{N}+C_{N}sinh\beta_{N}l_{N}+D_{N}cosh\beta_{N}l_{N}\right)=0 \end{array}$$

A characteristic matrix *G* of size (4Nx4N) is derived from mode shape solutions equations with applied boundary conditions. Solutions to $\beta_{n}$ are obtained by multiplying *G* with a vector of mode shape coefficients *P* and equating it to zero, at resonance frequencies (Equation (Equation 25)).
(25)$$G_{4Nx4N}P_{1x4n}=0$$
where
(26)$$P=[A_{1},B_{1},C_{1},D_{1},A_{2},B_{2},C_{2},D_{2},\cdots \cdots \cdots .,A_{N},B_{N},C_{N},D_{N}]$$

To obtain non-trivial solutions to Equation (Equation 25), varying values of β in small steps spanning a desired range are applied to matrix *G* and by finding zeros for its determinant, the natural frequencies can be found (see Equation (Equation 27)).
(27)det[G(β)]=0

The free vibration natural frequencies are then obtained from:(28)$$\omega_{r}^{2}={\left(\beta^{r}\right)}^{4}\frac{{\left(EI\right)}_{1}}{m_{1}}={\left(\beta_{n}^{r}\right)}^{4}\frac{{\left(EI\right)}_{n}}{m_{n}}$$
where $\beta_{r}$ are solution and $\omega_{r}$ is the $r^{th}$ natural frequency.

The system of ODE developed from Equations (Equation 15)–(Equation 28) is solved in MATLAB using the values given in Table 1 and the first three natural frequencies of the free vibration of the system obtained.

#### 2.1.2. Forced Vibration Analysis

Forced vibration formulation of the system was developed by assuming the application of a sinusoidal driving voltage to the electrodes of the piezoelectric patch, with a frequency close to the previously obtained natural frequencies ($\omega_{n}$) of the beam. Using results from the modal analysis, the solution of Equation (Equation 15) is assumed to be a linear combination of the normal modes of the beam, i.e.,
(29)$$w\left(x,t\right)=\sum_{r=1}^{\propto }\phi^{\left(r\right)}\left(x\right)q^{\left(r\right)}\left(t\right)$$
where $\phi^{r}$ is the $r^{th}$ mode shape function found by solving Equation (Equation 15) using the boundary conditions (see Equations (Equation 19), (Equation 21), (Equation 23)), and $q^{r}\left(t\right)$ are generalized coordinates or modal participation coefficients. Considering Equation (Equation 12) with the damping coefficient, it is written as:(30)$$\sum_{r=1}^{\propto }\left\{\frac{d^{2}}{dx^{2}}\left[E\left(x\right)I\left(x\right)\frac{d^{2}\phi^{\left(r\right)}\left(x\right)}{dx^{2}}\right]q^{\left(r\right)}\left(t\right)+c\left(x\right)\phi^{\left(r\right)}\left(x\right){\dot{q}}^{\left(r\right)}\left(t\right)+m\left(x\right)\phi^{\left(r\right)}\left(x\right){\ddot{q}}^{\left(r\right)}\left(t\right)\right\}=F_{P}\left(x,t\right)$$

By multiplying the $s^{th}$ mode shape on each side of the equation, the term with stiffness (EI) can be isolated from the equation.
(31)$$\int_{l_{0}}^{l_{N}}\left\{\sum_{r=1}^{\propto }\left(\frac{d^{2}}{dx^{2}}\left[E\left(x\right)I\left(x\right)\frac{d^{2}\phi^{\left(r\right)}\left(x\right)}{dx^{2}}\right]\phi^{\left(s\right)}\left(x\right)q^{\left(r\right)}\left(t\right)+c\left(x\right)\phi^{\left(s\right)}\left(x\right)\phi^{\left(r\right)}\left(x\right){\dot{q}}^{\left(r\right)}\left(t\right)+m\left(x\right)\phi^{\left(r\right)}\left(x\right)\phi^{\left(s\right)}\left(x\right){\ddot{q}}^{\left(r\right)}\left(t\right)\right)\right\}=\int_{l_{0}}^{l_{N}}F_{P}\left(x,t\right)\phi^{\left(s\right)}\left(x\right)dx=0$$

By isolating and evaluating the term with stiffness alone, Equation (Equation 31) is simplified into:(32)$$\int_{l_{0}}^{l_{N}}\left(\frac{d^{2}}{dx^{2}}\left[E\left(x\right)I\left(x\right)\frac{d^{2}\phi^{\left(r\right)}\left(x\right)}{dx^{2}}\right]\phi^{\left(s\right)}\left(x\right)q^{\left(r\right)}\left(t\right)\right)dx=\sum_{n=1}^{N}\int_{l_{0}}^{l_{N}}\left[{\left(EI\right)}_{n}\frac{d^{4}\phi_{n}^{\left(r\right)}\left(x\right)}{dx^{4}}\phi_{n}^{\left(s\right)}\left(x\right)q^{\left(r\right)}\left(t\right)\right]dx=\sum_{n=1}^{N}\left[\int_{l_{n=1}}^{l_{n}}\omega_{r}^{2}m_{n}\phi_{n}^{\left(r\right)}\phi_{n}^{\left(s\right)}\left(x\right)q^{\left(r\right)}\left(t\right)dx\right]$$

The term with stiffness alone is, therefore, simplified and written as:(33)$$\omega_{r}^{2}q^{\left(r\right)}\left(t\right)\int_{l_{0}}^{l_{N}}m\left(x\right)\phi^{\left(r\right)}\left(x\right)\phi^{\left(s\right)}\left(x\right)dx$$

By inserting Equation (Equation 33) into (Equation 31) and re-arranging, the forced vibration equation becomes
(34)$${\ddot{q}}^{\left(rt\right)}\left(t\right)+\sum_{s=1}^{\propto }\left\{c\left(x\right)\phi^{\left(r\right)}\left(x\right)\phi^{\left(s\right)}\left(x\right)dx\right\}+\omega_{r}^{2}q^{\left(r\right)}\left(t\right)=\int_{l_{0}}^{l_{N}}F_{p}\left(x,t\right)\phi^{\left(r\right)}\left(x\right)dx$$

Equation (Equation 34) is therefore simplified into
(35)$${\ddot{q}}^{\left(rt\right)}\left(t\right)+\sum_{s=1}^{\propto }\left\{c_{rs}\dot{q}\left(t\right)\right\}+\omega_{r}^{2}q^{\left(r\right)}=F^{\left(r\right)}\left(t\right)$$
where
(36)$$c_{rs}=\int_{l_{0}}^{l_{N}}c\left(x\right)\phi^{\left(r\right)}\left(x\right)\phi^{\left(s\right)}\left(x\right)dx,\;F^{\left(r\right)}\left(t\right)=F_{p}\left(x,t\right)\phi^{\left(r\right)}\left(x\right)dx$$

The damping coefficient (*c*) of the cantilever was derived from the combination of both air damping along the length of the cantilever, and material damping, with AlN considered the only contributor due to material damping of the polysilicon, used as the substrate, being considered negligible for these simulations, owing to its low intrinsic damping [52]. Air damping is calculated using kokubun et al’s model [53], based on stake’s law to derive the air damping coefficient. Experimental data points carried out by Hartono in [54] show that the air damping coefficient is given by:(37)$$C_{Air}=\sum_{0}^{L}b\left(x\right)\sqrt{8\pi \frac{m(x)}{k_{b}T}P}$$
where $b\left(x\right),m\left(x\right),T,k_{b}$, and *P* are the variable beam width, variable beam sectional mass, temperature, Boltzmann’s constant, and atmospheric pressure in Pascals, respectively, at which the damping is evaluated. Aluminum nitride (piezoelectric material) is considered the sole contributor to material damping. The material damping is determined as shown in Equation (Equation 38), where *B* is a material damping constant and has been determined to be the ratio of the material viscosity to the material Young’s modulus. For aluminum nitride, *B* has been determined to be $3.84\times 10^{-14}$ [55].
(38)$$C_{mat}=\frac{B_{AlN}}{\pi \omega_{n}}$$

The total damping *c* of the system is the total of the material and the air damping as shown in Equation (Equation 26).
(39)$$c=C_{mat}+C_{Air}$$

The ODE resulting from Equation (Equation 35) was transformed into the state space representation and solved in MATLAB for the first three natural frequencies.

### 2.2. FEM Analysis in ANSYS

A geometric model of the PD-MC with dimensions used in the numerical analysis was made using Ansys parametric design language (APDL). Two bodies were designed (the elastic beam body and the piezoelectric body) with the respective materials defined. The resultant volumes were glued together with the piezoelectric volume at the clamped end of the beam. Boundary conditions were applied to the resulting geometric model and imported to the ANSYS Workbench 19.2 modal module, where a mesh was applied and corresponding modal resonance frequencies were obtained. The result from the modal module was then connected to the harmonic response module and resonance frequency was obtained using the largest displacement of the node at the midpoint of the free end edge. Since the Euler–Bernoulli beam theory was applied, resonance frequencies due to transverse vibration without shear deformations were the only ones considered.

### 2.3. Equivalent Circuit Analysis

The piezoelectrically driven microcantilever was considered to be driven electrically by applying a known sinusoidal voltage at a frequency spanning its resonance frequency. The response obtained is in the form of complex impedance and the phase output, from which the resonance frequency can be deduced. The electrical equivalent model of the piezoelectric activated cantilever was, therefore, derived to allow for the analysis of the complete electromechanical system interactions for all modes of interest. The equivalent circuit model is then solved numerically, consequent to the determination of the lumped parameters of the equivalent circuit model.

The used model for the electrical response is the Butterworth–Van Dyke model [49], with lumped element parameters, as shown in Figure 2, where the lumped element parameters of the equivalent circuit are the inductance ( $L_{m}$), resistance ($R_{m}$), and capacitance ($C_{m}$), which form the motional arm of the circuit. Parallel capacitance $C_{0}$ forms the parallel arm of the circuit and is equivalent to the static capacitance of the piezoelectric patch between the electrodes.

The equivalent circuit modeling was evaluated using the outputs of the analytical model. The static capacitance $C_{0}$ can be defined according to the capacitances constitutive Equation (Equation 40) [56,57] where $A_{P},t_{p},\epsilon_{33}^{T},d_{31}$ and $S_{11}^{E}$ are the areas covered by the piezoelectric patch, thickness of the piezoelectric patch, dielectric constant under constant stress, piezoelectric strain coefficient, and the elastic compliance under the constant electric field, respectively.
(40)$$C_{0}=\frac{A_{p}}{t_{p}}\left(\epsilon_{33}^{T}-\frac{d_{31}^{2}}{S_{11}^{E}}\right)$$

$R_{m}$ is dubbed the motional resistance. According to circuit theory, $R_{m}$ is reduced to its minimum value at the resonant frequency of $\omega_{n}$ and is inversely proportional to the quality factor (Q) [57], which can be evaluated as in Equation (Equation 41). Using the solutions to the mode shape equations $\phi_{n}\left(x\right)$ resulting from analytical vibration analysis, the $R_{m}$ is defined as in Equation (Equation 43).
(41)$$Q=\frac{1}{2\zeta }$$
(42)$$R_{m}={\left(\frac{s_{11}^{E}}{wp*d_{31}}\right)}^{2}(\frac{2t_{p}m_{n}^{e}\left(E_{b}t_{b}+E_{p}t_{p}\right)\left(E_{p}t_{p}+E_{p}t_{b}\right)}{E_{b}t_{b}t_{p}\phi {\overset{´}{}}_{n}\left(l_{1}\right)Q{\left(t_{p}+t_{b}\right)}^{2}\int_{0}^{l_{1}}\frac{\partial^{2}\phi_{n}\left(x\right)}{\partial x^{2}}dx})$$
where ζ is the damping coefficient, $m_{e}$ is the effective mass of the microcantilever, and ${\dot{\phi }}_{n}\left(l_{1}\right)$ is the spatial derivative of the deflection of the cantilever at position $l_{1}$ (the endpoint of the piezoelectric element patch). The motional inductance $L_{m}$ and motional capacitance $C_{m}$ can be defined in terms of $R_{m}$ [49] as follows:(43)$$L_{m}=\frac{R_{m}Q}{w_{0}}$$
(44)$$C_{m}=\frac{1}{\left(w_{0}R_{m}Q\right)}$$

The equivalent resistance and reactance are expressed as
(45)$$R_{e}=\frac{R_{m}/\left(\omega_{n}^{2}C_{0}^{2}\right)}{R_{m}^{2}+{\left(\omega_{n}L_{m}-\frac{1}{\omega_{n}C_{m}}-\frac{1}{\omega_{n}C_{0}}\right)}^{2}}$$
(46)$$X_{e}=\frac{\frac{1}{\omega_{n}C_{0}}\left[R_{m}^{2}+\left(\omega_{n}L_{m}-\frac{1}{\omega_{n}C_{m}}\right)+\left(\omega_{n}L_{m}-\frac{1}{\omega_{n}C_{m}}-\frac{1}{\omega_{n}C_{0}}\right)\right]}{R_{m}^{2}+{\left(\omega_{n}L_{m}-\frac{1}{\omega_{n}C_{m}}-\frac{1}{\omega_{n}C_{0}}\right)}^{2}}$$

Moreover, the complex impedance is evaluated as:(47)$$Z=R_{e}+jX_{e}$$

The electronic response in the form of impedance and phase of the cantilever was evaluated using MATLAB, with the basis on the simulated cantilever deflections from the forced vibration analysis. A range of masses from 0–200 μg was added to the free end of the cantilever, to simulate the change in resonance frequency, and the changes in the impedance and phase response curves with the mass are recorded.

### 2.4. Experimental Verification

A microcantilever provided by Fraunhofer ISIT with the same dimensions as those used in the analytical model formulation and with a measured resonance frequency of 10.02 kHz was used. The impedance and phase response were measured using an Agilent 42941A precision impedance analyzer [58] with a measurement tolerance of ±3.0 Hz. The impedance analyzer was set to use the Butterworth–Van Dyke equivalent circuit model, with a driving source voltage of 1 V. The impedance analyzer was set to use the Butterworth–Van Dyke equivalent circuit model, with a driving source voltage of 1 V. For ease of reading, we use the term “calibration” to mean the application of known masses to obtain a resonance frequency response that is later used to develop the response polynomial.

Two sets of calibrations were done, one with calibration generic calibration masses and the other with measured masses of the binder solution. The provided binder solution had strong adhesive properties with glue-like characteristics to enable it to stick to the microcantilever surface. The response profiles from the two sets of functionalization masses were, therefore, expected to be different owing to the difference in surface stress each category of mass induces on the microcantilever beam.

For the first set of calibration, henceforth referred to as “calibration set-1”, the microcantilever was incrementally dozed with 0.01 μL up to a homogeneous solution of density 2.1 g/mL using a Hamilton Gastight Syringe, mounted on the OCA 15 EC, optical contact angle meter from data physics [59]. The maximum mass used for calibration was 271.96 μg. The corresponding mass to each dozing was calculated, and the resulting resonance frequency was extracted from the impedance analyzer (see Figure 3 ). The performance of the obtained polynomial was tested by adding three different volumes of 98% purity 1-dodecanethiol obtained from Sigma Aldrich, equating to 169 μg, 338 μg, and 507 μg, while recording the resulting resonance frequency.

For the second set of calibration, henceforth referred to as “calibration set-2”, the microcantilever was cleaned in an ultrasonic bath of isopropanol and left for 1 h to dry, prior to each functionalization. The clean microcantilever was then functionalized by drop-casting with the binder solution of 1,5-pentanediamine (cadaverine), provided by AmiNIC ApS. The same setup for the micro-dosing set-up as that in Figure 4. For each functionalization, the mass of the binder on the microcantilever was measured by thermogravimetric analysis, using a TGA-550, from Waters/Thermogravimetric analyzers Instruments [60], with a measurement tolerance of ±0.1 μg. The measured masses were 2.1 μg, 7.3 μg, 13.3 μg, and 18.6 μg. The resonance frequency response of the complete setup was measured and recorded for each mass of the binder solution. The resulting data points of resonance frequency response to the added mass from both sets of calibration were then fit to respective polynomials using Polyfit and Polyval functions in MATLAB, and the values to the α, β, γ, and λ coefficients to Equation (Equation 2) were obtained. The norm of residuals was used to determine the goodness of fit of the polynomial, with a lower norm signifying a better fit. The performance of the resulting polynomial was tested by adding 3.6 μg, 3.1 μg, and 3.1 μg of the binder solution to the cleaned microcantilever. The resonance frequency for each mass was recorded and estimated using the polynomial.

For the estimation of the mass of adsorbed cadaverine, a functionalized microcantilever with 3.4 μg of the binder solution was exposed to incremental volumes of 0.2 mL, 0.3 mL, and 0.4 mL of 95% pure cadaverine, purchased from Sigma Aldrich, in a closed 750 mL container. The concentrations of the cadaverine for each volume, in parts per million-volume ($ppm_{v}$), were calculated to be 0.0551,0.0827,0.1103$ppm_{v}$ using Equation (Equation 48) as described in [61].
(48)$$C_{o}\left(ppm_{v}\right)=\frac{22.4\rho TV^{{}^{\prime }}}{273.15MV}\times 1000$$
where $C_{o}$ is the sample concentration, ρ is the density of the sample, $V^{{}^{\prime }}$ is the volume of sample put in the container, *T* is the temperature in Kelvins, *M* is the molecular weight of the sample, and *V* is the volume of the chamber in Litres.

## 3. Results

### 3.1. Analytical Vibration Analysis

Table 2 shows the resonance frequencies obtained from the numerical analysis by finding solutions to Equation (Equation 25), in comparison to those obtained from the simulation in ANSYS. The error between the simulation results and the analytical modal analysis results is also shown.

### 3.2. Simulation of the Resonance Frequency Response of Microcantilever to the Added Mass

Consequent to the forced vibration analysis, an analysis of the change in resonance frequency with added mass, and the numerical analysis of the corresponding electrical impedance and phase response was performed by adding masses between 0 and 200 μg to the free end of the microcantilever model. For ease of the simulation, the homogeneous distribution of the masses on the cantilever surface was assumed.

Figure 5 shows the profile of the change of resonance frequency with mass added to the cantilever. There is a linear trend for very small masses, i.e., between 0 and 10 μg. The response, however, becomes non-linear with an increase in mass, in a polynomial-like trend. Based on our hypothesis, we see a possibility of parameterizing this response trend into a polynomial, to provide a basis on which mass adsorbed on the cantilever can be determined.

### 3.3. Experimental Results

A plot of the actuated microcantilever resonance frequency response from calibration set-1 (where generic masses are used) shows a reduction in the resonance frequency with the increase in the micro-doses of known masses (see Figure 6). A linear connection of the resonance frequency data points shows varying gradients between each point (dotted line). By fitting the resonance frequency response data points into a polynomial using Polyfit and Polyval functions in MATLAB, a third-order polynomial was obtained. The resulting norm of residuals for the polynomial fit was 0.16309 (see Figure 6 solid line). Table 3 shows the resonance frequencies read from the equipment consequent test masses added, in comparison with estimated masses based on the previously obtained resonance frequency to the added mass response polynomial. The table also shows the estimation of added masses outside the calibration range.

Using the resulting polynomial to estimate the mass when a test mass of 69 μg of the test mass is used, shows an estimated mass of 170 μg, giving a difference of 1.0 μg (Figure 7). Moreover, 169 μg is in the range of calibration masses, ranging from 0 to 271.96 μg. Estimating masses outside of the calibration range shows a significant difference in the masses estimated and known masses added. The difference increases with the increase in mass added.

Figure 8 shows the response polynomial resulting from the use of known masses of functionalization binder solution. The resulting norm of residuals for the polynomial fit was 0.056417. The results from testing the resultant response polynomial with measured masses of 3.1 μg, 3.4 μg, and 4.0 μg of the binder solution show estimated masses of 3.3 μg, 3.6 μg, and 4.07 μg, respectively. Table 4 and Figure 9 show the estimations of the added masses using the resulting polynomial.

Figure 10 shows the comparison between the parameterized resonance frequency to the added mass response resultant polynomial from calibration set-1 using known masses, and calibration set-2 using the binder solution for the first 20 μg. This is because calibration set-2 was done with a maximum mass of 20 μg. The graph also shows that the analytically obtained resonance frequency response to the added mass response obtained analytically is lower than those obtained from both parameterized polynomial responses from calibration set-1 and calibration set-2. The responses from the two parameterized polynomials also show significant differences in the rate of the resonance frequency drop-off with mass added. The response polynomial from calibration set-1 has a lower drop-off rate compared to the response polynomial from calibration set-2.

Figure 11 shows the resonance frequencies obtained when a functionalized cantilever with 3.4 μg of the binder was exposed to 0.0551,0.0827,0.1103
$m_{v}$ calculated concentrations of the cadaverine were 8.9618 kHz, 8.8833 kHz, and 8.8520 kHz, respectively. These resonance frequencies translated to 4.7 μg, 5.7 μg, and 6.2 μg of cadaverine adsorbed (see Table 5).

## 4. Discussion

The analytical simulation of the micro-cantilever resonance frequency to the added mass response showed a non-linear polynomial-like trend with an increase in the mass. The results confirmed the possibility of applying a polynomial fit to the actuated microcantilever resonance frequency to the added mass response. The rate of change of the resonance frequency response with the known added mass in the polynomial realized from calibration set-1 is lower than that realized from calibration set-2 and the simulated response. This is attributed to the differences in damping accruing from each setup. The analytical model used estimates the damping from predefined models of air and material damping, excluding non-classical sources that were not accounted for. There are other contributions to damping especially those originating from non-classical damping contributions, such as the coupling of the piezoelectric layer dynamics to the mechanical dynamics, for instance, the counter strain due to self-induced voltage [62], adsorption-induced damping [45], and shear deformations, especially for beams that are widely compared to the length [63]. There is, therefore, a risk of always over-estimating or under-estimating the damping coefficient. This phenomenon has been widely investigated and different methods of approximations were developed (although they are lacking in accuracy) [64].

Most of the piezoelectrically driven microcantilever sensor system damping is challenging to model accurately; therefore, it is not fully accounted for in most analytical and simulation models. By obtaining the response of a piezoelectrically driven microcantilever in a complete setup with actuating and readout equipment, it was possible to deduce the actual response of the system combined with all sources of noise, damping, and electrical losses. By parameterizing the resonance of the microcantilever resonance frequency to the added mass, a third-order polynomial for use in the estimation of the mass of the target gas was adsorbed.

The differences in parameterized response profiles from both calibration sets are attributed to the differences in adhesive properties of the two materials used for calibration. The binder solution used in calibration set-2 has glue-like sticky properties to allow for immobilization when applied to the microcantilever surface. This results in the introduction of surface stresses on the microcantilever, thereby damping the actuated vibrations. The damped actuated vibrations in turn result in damped resonance frequencies. This observation led to the deduction that the material used for calibration should have microcantilever surface adhesive properties similar to those of the material used in the sensing application.

On testing the performances of the two microcantilever response polynomials to added mass, it was found that each polynomial could only be used for the estimation of masses with similar adhesive properties to those of materials of known masses. Looking at the response polynomial developed from calibration set-1, a known mass of 169 μg of 1-dodecanethiol was estimated as 170 μg, with an error of 1 μg. Without regard for experimental errors, this can be considered accurate for masses in the order of tens of micrograms. The highest mass used in the calibration was 271 μg, which lies within the calibration range. An attempt to use the polynomial to estimate masses outside the calibration range was futile, with estimated masses and resonance frequencies significantly different from known test masses used in the experiment and resonance frequencies read from the equipment. This same result was observed when known masses of the binder solutions were used. The estimations of three differently known functionalization masses with sub-microgram differences were more accurate with an average error of 0.16 μg. This improvement in accuracy was attributed to the use of the same solutions as both calibration masses and test masses. For accurate use of the resulting response polynomials, the masses to be estimated should be within the range of masses used for calibration, and the material of mass under the test should have the same adhesive properties as the microcantilever surface, such as the masses used in calibration.

Consequent to the performance of the response polynomial from calibration set-2, it was then used to estimate the mass of the adsorbed cadaverine on a functionalized microcantilever in different cadaverine concentrations. There was a reduction in the microcantilever response resonance frequency, with an increase in concentration. The estimated masses were 4.7 μg, 5.7 μg, and 6.2 μg for the 0.0551, 0.0827, and 0.1103 $m_{v}$ calculated concentrations of cadaverine, with resonance response frequencies of 8.9618 kHz, 8.8833 kHz, and 8.8520 kHz, respectively. These results are withing the assumption that there is no desorption and that the interaction between the binder and adsorbed cadaverine molecules has a negligible effect on the viscosity of the binder. The change in the binder viscosity influences the overall surface stress, subsequently affecting the resulting resonance frequency. The estimated mass of the adsorbed cadaverine is in micrograms, which can be deemed high for a gas. However, it is important to note that synthetic cadaverine was used. The cadaverine has a molecular weight of 102.18 g/mol and a settling density of 870 kg/mol [65]. Despite the fact that the cadaverine is a volatile gas, it crystallizes when exposed to air and forms heavy particles. This made it possible to estimate the mass of the adsorbed cadaverine since the polynomial was developed using masses in the order of micrograms. For an accurate response in nanograms or picograms, it is imperative that known masses (in the order of target masses for the estimation) be used for developing the resonance frequency to the added mass response polynomial.

## 5. Conclusions

An analytical solution for the transverse vibration analysis of a stepped beam extendable to beams with desired structures and shapes was derived. A free and forced vibration analysis for a provided cantilever with known structures and dimensions was conducted. An Ansys simulation was carried out to verify the developed analytical model and they were proximate, especially for the first mode, which is of our interest. The resonance frequency for the first mode from the analytical solution was 10.553 kHz while that from the simulation in ANSYS was 10.553 kHz. An analytical resonance frequency to the mass added to the model was also conducted and a polynomial-like response profile of the microcantilever was obtained.

The system set-up of the piezoelectrically driven microcantilever, together with the driving and read-out equipment, was calibrated with known masses to obtain a resonance frequency to the added response. The response data points were then used as the bases (i.e., parameterizing the microcantilever response into a polynomial of the order of three), based on which unknown masses could be estimated. The approach was exclusively applicable to the estimation of masses within the range of masses used for calibration. It was also found that the accuracy of the resulting polynomial used for the estimation of adsorbed masses is highly dependent on the order of masses and adhesive properties of masses used for calibration.

It can be concluded that there is a possibility of using this approach to estimate the cadaverine adsorbed on a microcantilever. The performance of this approach is, however, limited by the known masses used for calibration.

The findings from this study are the bases on which the calibration of microcantilevers used for biogenic gas sensing can be done. This will allow for accurate estimation of the biogenic gas concentration in a sample, as derived from the mass of the gas adsorbed onto the functionalized cantilever.

## Figures and Tables

**Figure 1 sensors-23-01093-f001:**
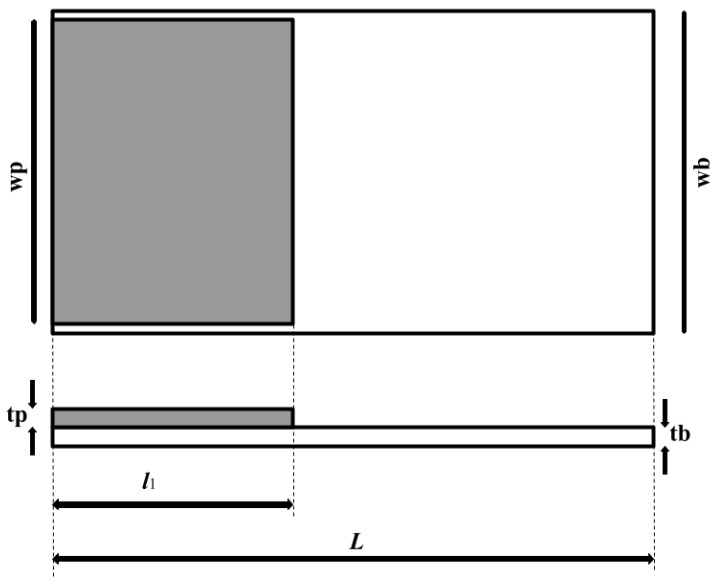
Chosen microcantilever beam design used for the baseline simulations.

**Figure 2 sensors-23-01093-f002:**
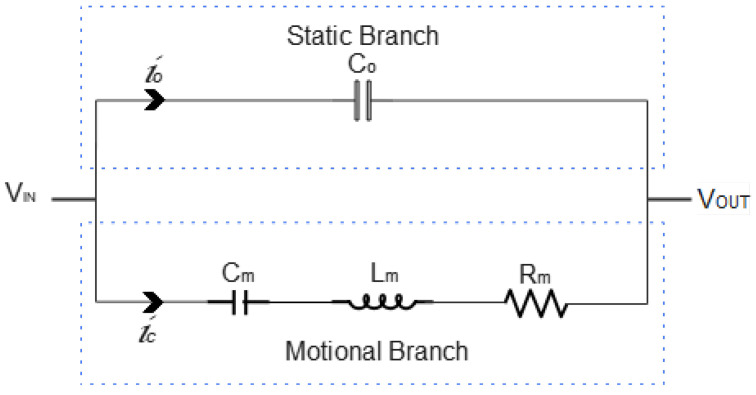
Lumped element for the Butterworth–Van Dyke model.

**Figure 3 sensors-23-01093-f003:**
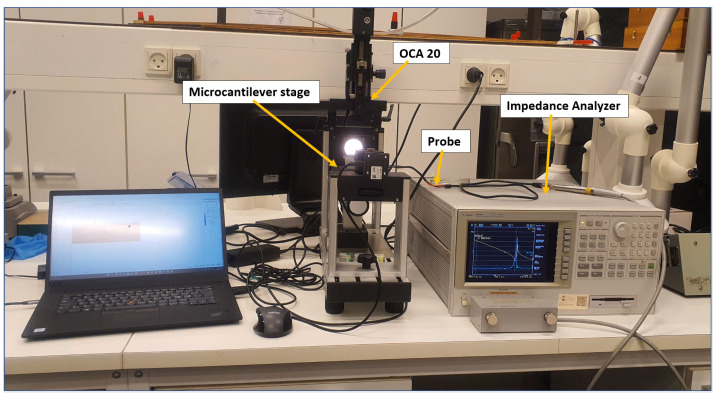
The set-up used for the calibrating microcantilever response to mass added.

**Figure 4 sensors-23-01093-f004:**
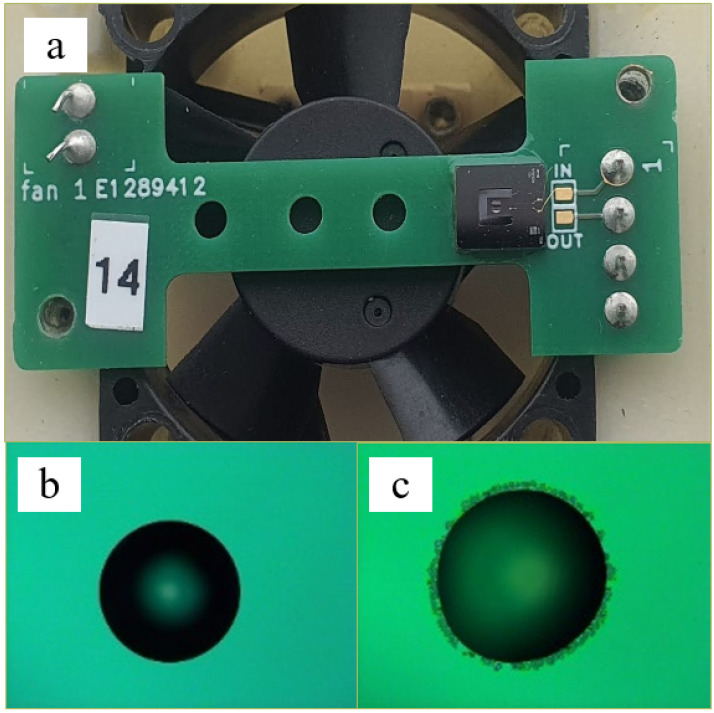
(**a**) Piezoelectrically driven microcantilever beam mounted on the PCB board. (**b**) Functionalization solution on the microcantilever before exposure to the cadaverine. (**c**) After exposure to the cadaverine.

**Figure 5 sensors-23-01093-f005:**
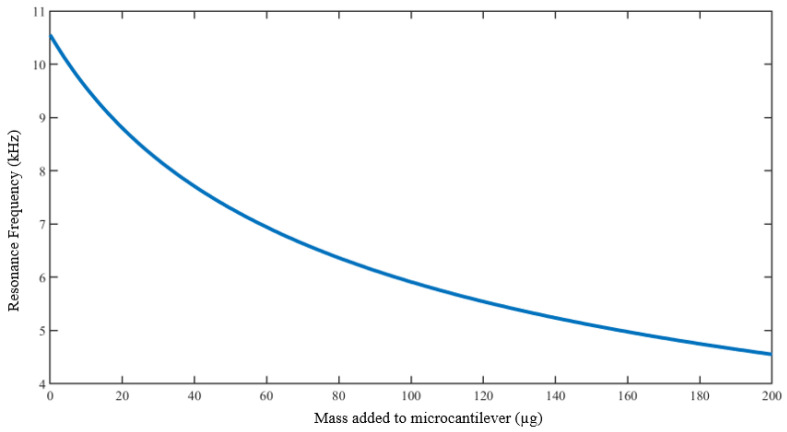
Simulated polynomial-like microcantilever resonance frequency response to the mass added, for the modeled piezoelectrically driven microcantilever beam.

**Figure 6 sensors-23-01093-f006:**
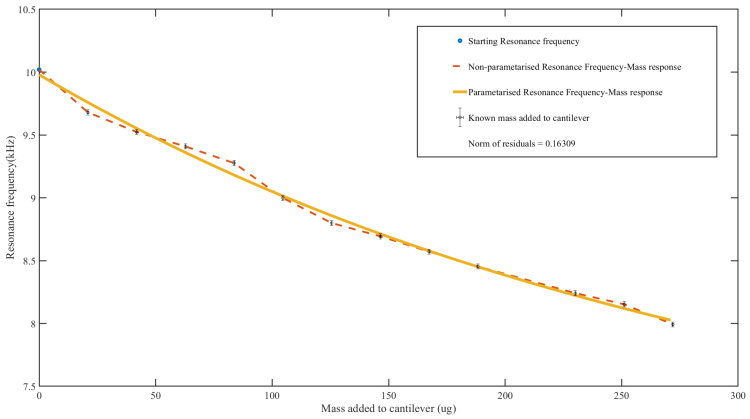
Parameterized resonance frequency to the added mass response calibration set-1 using generic masses.

**Figure 7 sensors-23-01093-f007:**
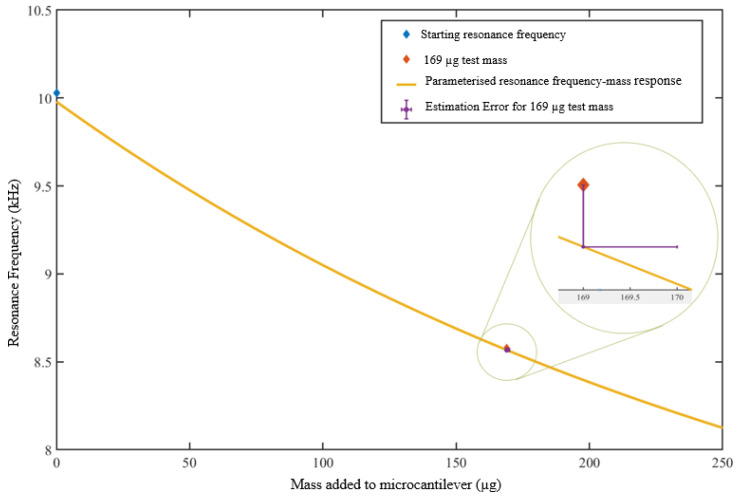
Estimation of 169 μg of added mass using the polynomial from calibration set-1.

**Figure 8 sensors-23-01093-f008:**
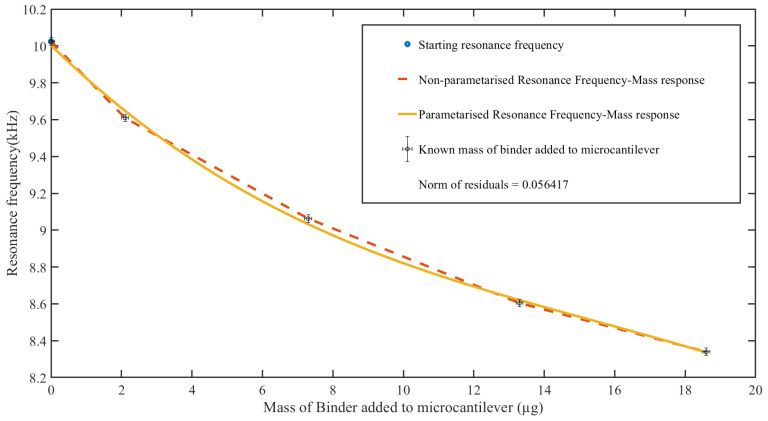
Calibration set-2 resonance frequency response to the added mass response of the microcantilever characterized into a polynomial (solid line).

**Figure 9 sensors-23-01093-f009:**
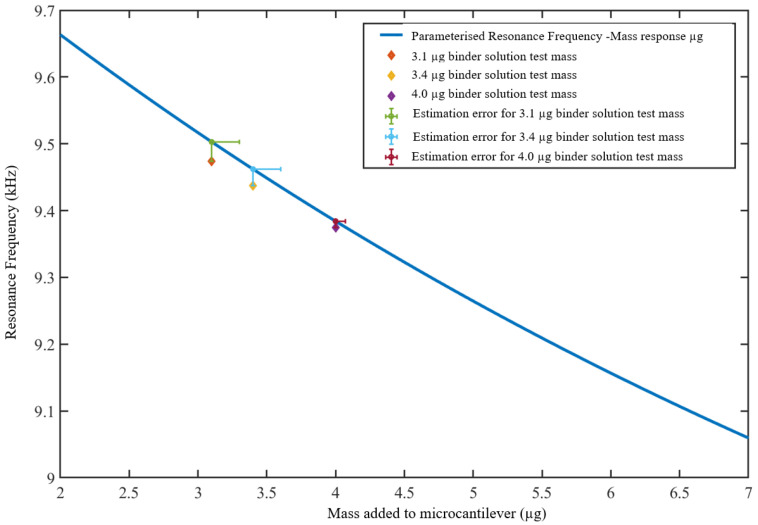
Estimation of the known test masses of the binder solution using the polynomial from calibration set-2.

**Figure 10 sensors-23-01093-f010:**
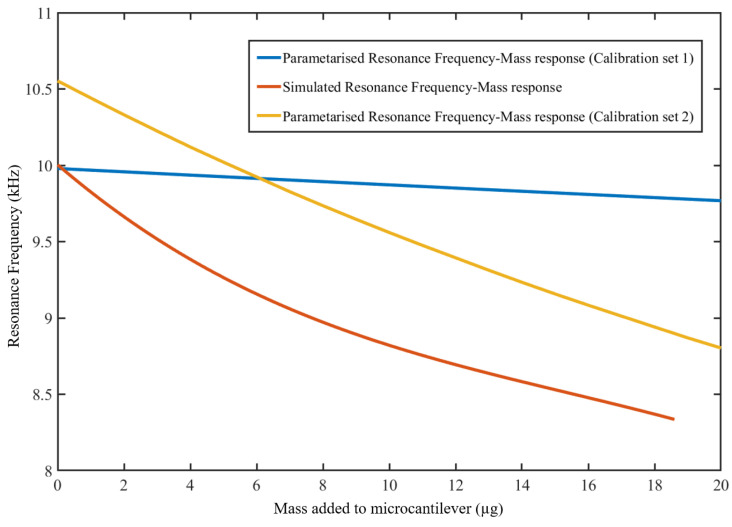
Comparison of simulated resonance frequency-mass response to parameterised response from calibration set-1 and calibration set-2.

**Figure 11 sensors-23-01093-f011:**
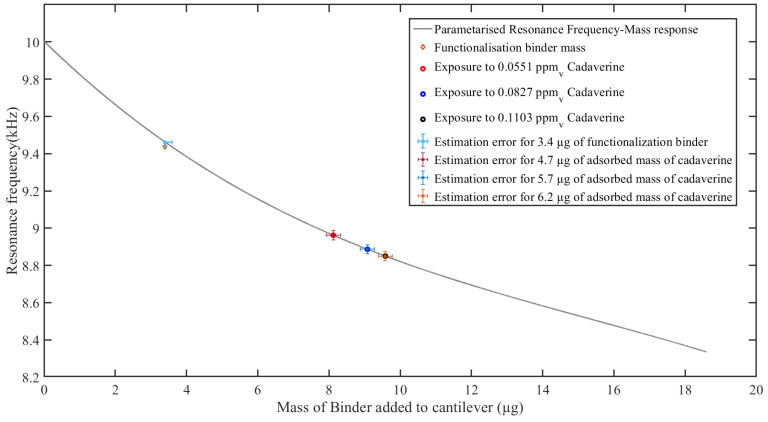
Estimation of the adsorbed mass on the microcantilever, using the polynomial from calibration set-2, after exposure to different concentrations of cadaverine.

**Table 1 sensors-23-01093-t001:** Values used to solve ODEs in MATLAB.

Property	Value	unit
l_1_	0.5	mm
l_2_	1.0	mm
w${}_{b}$	1.0	mm
w${}_{p}$	1.0	mm
tb	20.0	μm
tp	2.0	μm
E${}_{b}$	70.0	GPa
E${}_{P}$	302.0	Gpa
d_3_1	−1.9159 × 10^−12^	C/N
ɛ${}_{P}$	11.5 × 10^−12^	-
$\rho_{P}$	3260	Kg/m^3^
$\rho_{b}$	2270	Kg/m^3^
Vt	5	V

**Table 2 sensors-23-01093-t002:** Comparison of resonance frequencies of the first three cantilever transverse vibration modes.

Mode	Resonance Frequency (kHz) Analytical Approach	Resonance Frequency (kHz) Numerical Approach	Difference in (kHz)	Percentage Error
Mode 1	10.553	10.556	0.003	0.028
Mode 2	52.987	52.707	0.28	0.005
Mode 2	135.37	132.92	2.45	0.018

**Table 3 sensors-23-01093-t003:** Resonance frequency response to added mass test masses with expected resonance frequency from the parameterized response. The (*) shows values due to masses outside the calibration range.

Known Mass (μg)	Measured Resonance Frequency (kHz)	Deduced Adsorbed Mass Based on Polynomial (μg)	Difference in Known Mass and Deduced Mass	Expected Resonance Frequency to Fit Polynomial (kHz)	Mass Estimation Tolerance (%)
169	8.5774	170	1	8.5671	0.59
338 *	8.2645	222 *	116 *	7.7464	52.25 *
507 *	7.9992	277 *	230 *	7.1174	83.03 *

**Table 4 sensors-23-01093-t004:** Response of the second set calibration polynomials based on the binder solution with test masses of the binder solution.

Known Mass (μg)	Measured Resonance Frequency (kHz)	Deduced Adsorbed Mass Based on Polynomial (μg)	Difference in Known Mass and Deduced Mass	Expected Resonance Frequency to Fit Polynomial (kHz)	Mass Estimation Tolerance (%)
3.1	9.4742	3.3	0.2	9.5027	6.06
3.4	9.4372	3.6	0.2	9.4620	5.56
4.0	9.3746	4.07	0.07	9.3841	1.72

**Table 5 sensors-23-01093-t005:** Estimation of the mass of cadaverine adsorbed when the functionalized microcantilever with 3.4 μg of the binder is exposed to different concentrations of cadaverine.

Concentration of the Cadaverine (${\mathrm{ppm}}_{v}g$)	Measured Resonance Frequency (kHz)	Deduced Total on the Microcantilever Based on the Polynomial (μg)	Deduced Adsorbed Mass of Cadaverine (μg)
0.0551	8.9618	8.1	4.7
0.0827	8.8833	9.1	5.7
0.1103	8.8520	9.5	6.2

## Data Availability

Not applicable.

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
