# Peer review of "Gas Adsorption Response of Piezoelectrically Driven Microcantilever Beam Gas Sensors: Analytical, Numerical, and Experimental Characterizations"

_sensors, 2023, doi:10.3390/s23031093_

Round 1

Reviewer 1 Report

1. What is the importance of sensing 1,5-diaminopentane and not another type of element or gas?

2. Check the grammar. Section 3.3, line 4.

3. Could you calculate the error between the numerical model and the experimental one? If so, it would be convenient to put a small table where the error values are exposed.

4. According to the previous point, it would be good to include some graph of the error, as well as include more graphs in the results obtained.

Author Response

Dear reviewer,

Thanks for your comments. Here are our replies:

  1. What is the importance of sensing 1,5-diaminopentane and not another type of element or gas?

The introduction has been modified to show the reason for sensing 1,5-diaminopentane in our study.

  1. Check the grammar. Section 3.3, line 4

The grammatical error has been rectified.

  1. Could you calculate the error between the numerical model and the experimental one? If so, it would be convenient to put a small table where the error values are exposed.

The numerical model just and the experimental model has been calculated. 3 results showing resonance frequency for known mass of the binder have been added to Figure 5 in the result section.

  1. According to the previous point, it would be good to include some graph of the error, as well as include more graphs in the results obtained.

More graphs have been added to the result section.

Reviewer 2 Report

In order to ensure that the paper is of the highest quality, it is recommended that certain topics be clarified and certain improvements be made. These changes could include, but are not limited to, addressing unclear points, providing additional evidence to back up claims, and revising grammar and sentence structure.

2: error: predetermined

4: Euler-Bernoulli

18: low-cost

25: reference for 1,5-diaminopentane(Cadaverine)

26: there is increase??

29: rewrite

44: developed? Uppercase

68: an analytical

72: reference of the BVD model

Equation 19: results into mode coefficients??

 whose determinant is ZERO at Resonance frequencies??

 The physical meaning of Cm, Rm and Lm should be defined.

 166: Some photograph and more details of the cantilever should be introduced.

179: reference of the instrument

188: a brief description of the modal shapes should be introduced

189: obtained

Table 2: more details about the numerical approach with ANSYS should be introduced

The figure 5 is a measurement or a simulation result?

206: response can be…

The description of figure 6 is not correct

The results presented in figure 6 are obtained with only one measurement??. Have been tried a series of measurement? Any idea about the error or resolution of the measurement? More details should be included regarding this aspect…

220: Experimental results??

225: Some details should be added regarding the possible measurement of a gas. In that case is key the addition of the sensor resolution…

Author Response

Dear reviewer,

Thanks for your valuable comments. Here are our replies:

In order to ensure that the paper is of the highest quality, it is recommended that certain topics be clarified and certain improvements be made. These changes could include, but are not limited to, addressing unclear points, providing additional evidence to back up claims, and revising grammar and sentence structure.

The relevant topics have been clarified and improvements to the introduction made. The grammar and sentence structure has also been extensively revised. More results have been included in the result section and references to claims made.

2: error: predetermined

The whole abstract has been revised and improved, and the grammar corrected.

4: Euler-Bernoulli

The error has been corrected.

18: low-cost

The claim has been removed since it is not an objective of our study.

 25: reference for 1,5-diaminopentane(Cadaverine)

References about 1,5-diaminopentane have been added.

26: there is increase??

The grammar has been corrected.

29: rewrite

The Introduction has been revised and rewritten.

44: developed? Uppercase

The Introduction has been revised and rewritten.

68: an analytical

The Introduction has been revised and rewritten.

72: reference of the BVD model

Reference to the BVD have been provided.

Equation 19: results into mode coefficients??

The explanation has been corrected.

 whose determinant is ZERO at Resonance frequencies??

The explanation has been corrected.

 The physical meaning of Cm, Rm and Lm should be defined.

A description of each variable has been provided.

 166: Some photograph and more details of the cantilever should be introduced.

Photographs  of the cantilever before and after functionalization has been included.

179: reference of the instrument

References to all the instruments used have been provided

188: a brief description of the modal shapes should be introduced

To avoid unnecessary results in the article, these have been omitted, corresponding resonance frequencies are included.

189: obtained

The spelling and grammar have been corrected.

Table 2: more details about the numerical approach with ANSYS should be introduced

A brief account of how ANSYS was used for the NUMERICAL analysis has been included. It has also been emphasized that it was only used to verify the analytical model.

The figure 5 is a measurement or a simulation result?

Figure 5 is a simulation result and has now been clearly explained in the results.

206: response can be…

The Grammar has been corrected.

The description of figure 6 is not correct

The description for Figure 6 has been corrected.

The results presented in figure 6 are obtained with only one measurement??. Have been tried a series of measurement? Any idea about the error or resolution of the measurement? More details should be included regarding this aspect…

More results for 2 more points have been included in the graph. The error between obtained and anticipated readings has also been included.

220: Experimental results??

The paragraph has been corrected.

225: Some details should be added regarding the possible measurement of a gas. In that case is key the addition of the sensor resolution…

There is a subsequent study with the aim of correcting for non-linearities by using a feedback controller. This study will provide more accurate results that we can depend on to give a resolution and a Limit of Detection for our sensor system. We are currently not able to provide a reliable resolution.

Reviewer 3 Report

First of all, it is difficult to find novelty in this paper. 

Also, the paper's data is very small, so it seems difficult to be of great help to readers. 

Figs. 4 and 5 are too general graphs, and only Fig. 6 is relevant to this study. 

I think it is difficult for this paper to be published in this journal.

Author Response

Dear reviewer,

Thanks for your comments. It might be that the structure of the paper did not make it clear enough. We have now modified the structure and hope you can find it relevant.